# Clinical Phenotypes and Predictors of Remission in Primary Membranous Nephropathy

**DOI:** 10.3390/jcm10122624

**Published:** 2021-06-15

**Authors:** Roxana Jurubiță, Bogdan Obrișcă, Bogdan Sorohan, Camelia Achim, Georgia Elena Micu, Gabriel Mircescu, Gener Ismail

**Affiliations:** 1Department of Nephrology, Fundeni Clinical Institute, 022328 Bucharest, Romania; roxana.jurubita83@gmail.com (R.J.); bogdan.sorohan@yahoo.com (B.S.); camelia_ailemac@yahoo.com (C.A.); elenageorgia.micu@gmail.com (G.E.M.); gener.ismail@umfcd.ro (G.I.); 2Department of Nephrology, “Carol Davila” University of Medicine and Pharmacy, 020021 Bucharest, Romania; gmircescu@hotmail.com; 3Department of Nephrology, “Dr. Carol Davila” Teaching Hospital of Nephrology, 010731 Bucharest, Romania

**Keywords:** primary membranous nephropathy, anti-PLA2R antibodies, proteinuria, clinical remission, immunosuppressive treatment, spontaneous remission

## Abstract

(1) Background: We sought to investigate the clinical outcome and to identify the independent predictors of clinical remission in a prospectively followed cohort of patients with primary membranous nephropathy (pMN). (2) Methods: We conducted a prospective, observational, non-interventional study that included 65 consecutive patients diagnosed with pMN between January 2015 and December 2019 at our department and followed for at least 24 months. The primary outcomes evaluated during the follow-up period were the occurrence of immunological and clinical remission (either complete or partial remission). Univariate and multivariate Cox proportional hazard regression analyses were performed to identify independent predictors of clinical remission. (3) Results: In the study cohort, 13 patients had a PLA2R-negative pMN, while, of those with PLA2R-associated pMN, 27 patients had a low anti-PLA2R antibody titer (<200 RU/mL), and 25 patients had a high anti-PLA2R antibody titer at baseline (≥200 RU/mL). The clinical outcome was better in patients with PLA2R-negative pMN compared to patients with PLA2R-positive pMN. These patients had a higher percentage of complete remissions (46.2%, compared to 33.3% in those with low anti-PLA2R antibody titer or 24% in those with high anti-PLA2R antibody titer), a faster decline of 24 h proteinuria and lower time to complete remission. In multivariate Cox regression analysis, patients with PLA2R-negative pMN had a 3.1-fold and a 2.87-fold higher chance for achieving a complete or partial remission compared to patients with high anti-PLA2R antibody titer or to all PLA2R-positive patients, respectively. Additionally, patients with a baseline 24 h proteinuria of less than 8 g/day and with an immunological remission at 24 months had a 2.4-fold (HR, 2.4; 95%CI, 1.19–4.8) and a 2.2-fold (HR, 2.26; 95%CI, 1.05–4.87), respectively, higher chance of achieving a clinical response. By contrary, renal function at diagnosis, type of therapeutic intervention or anti-PLA2R antibody titer did not predict the occurrence of clinical remission. (4) Conclusions: We identified a different clinical phenotype between PLA2R-positive and PLA2R-negative pMN. Additionally, we have shown that baseline proteinuria seems to be a more important predictor of clinical outcome than anti-PLA2R-ab titer.

## 1. Introduction

Despite the increasing prevalence of focal and segmental glomerulosclerosis in certain subpopulations, primary membranous nephropathy (pMN) remains an important cause of primary nephrotic syndrome (NS) in nondiabetic adults, affecting mostly Caucasians with ages between 30 and 50 years [1].

Primary membranous nephropathy is an autoimmune glomerulopathy determined by autoantibodies targeting antigens expressed on the podocyte’s surface [2]. Since the landmark study in 2009 describing the first human target autoantigen, M-type phospholipase A2 receptor 1 (PLA2R) [3], the landscape of pMN has significantly changed with the description of several new putative “antigens” (THSD7A, exostosin 1/exostosin 2, NELL1, semaphorin 3B, protocadherin 7) [4,5,6,7,8]. Accordingly, it becomes evident that each distinct antigen might drive a different clinical phenotype in the context of a different pathophysiological phenomenon [9]. Nonetheless, anti-PLA2R antibodies account for up to 50–80% of pMN cases [2]. The diagnostic value of serum anti-PLA2R antibodies (anti-PLA2R-ab) has been evaluated in several studies with anti-PLA2R-ab showing a good diagnostic accuracy to differentiate pMN from secondary MN or other non-MN patterns of glomerular injury [10]. In addition to their diagnostic role, anti-PLA2R-ab correlate strongly with disease activity and emerged as an important prognostic biomarker [11,12,13,14,15,16,17,18,19,20]. A recent meta-analysis showed that patients with anti-PLA2R-ab had a poor clinical outcome with a lower clinical remission rate and a higher risk of renal failure [21]. Until the recently described antigens are fully validated and each distinct antigen-associated MN is clearly defined, the PLA2R-negative pMN should acknowledged as a distinct entity with a different clinical phenotype and outcome [18]. Moreover, the most important predictors of clinical outcome in pMN remain uncertain [21].

Accordingly, we sought to investigate the clinical outcome and to identify the independent predictors of clinical remission in a prospectively followed cohort of patients with pMN.

## 2. Materials and Methods

### 2.1. Study Design and Population

We conducted a prospective, observational, noninterventional study that included 65 consecutive patients diagnosed with pMN between January 2015 and December 2019 at our department.

Primary MN was diagnosed by percutaneous kidney biopsy. All biopsies were examined under immunofluorescence, light and electron microscopy by an experienced pathologist. In addition to the characteristic histopathologic features of MN, the biopsies were also assessed for features suggestive of secondary MN: immunofluorescence with full-house staining, vascular or tubular basement membrane staining, mesangial staining; light microscopy with mesangial and endocapillary hypercellularity; electron microscopy with subendothelial or mesangial deposits, vascular or tubular basement membrane deposits, tubuloreticular inclusions [22]. Additionally, all patients underwent a systematic screening for secondary causes of pMN such as autoimmune disorders screening, hepatitis serology, age-appropriate malignancy screening, monoclonal gammopathy evaluation and medication history [22]. Exclusion criteria were: age < 18 years old, patients with prior immunosuppression (IS), patients with features suggestive of secondary MN, patients with a follow-up period of less than 24 months, patients with incomplete data or with missed visits and patients with a positive PLA2R serology but without a confirmatory kidney biopsy. Of the 92 patients diagnosed with pMN in our clinic, 27 were excluded (15 with a follow-up period of less than 24 months and 12 patients with inadequate data during follow-up; of these, 10 had potential secondary causes of NS, leaving a final cohort of 65 patients).

The study was conducted with the provisions of the Declaration of Helsinki, and the protocol was approved by the local ethics committee (The Ethics Council of Fundeni Clinical Institute, Registration number: 8851). All patients provided a written informed consent prior to study enrollment.

### 2.2. Treatment

Patients were treated in accordance with current KDIGO guidelines and the most recent randomized trials [22,23,24,25,26]. The treatment was conducted at the discretion of the attending physician without any intervention. Patients included in the study were either IS-naïve or received either a cyclophosphamide-based, calcineurin inhibitor-based or a rituximab-based regimen. Patients with a cyclophosphamide-based regimen received the classical Ponticelli cyclic regimen, consisting of three consecutive cycles lasting 2 months each (for a total of 6 months), where steroids were alternated with cyclophosphamide every other month [26,27]. Patients in the calcineurin-inhibitors (CNI) group received cyclosporine at an oral dose of 3.5–5 mg per kilogram per day, adjusted in order to obtain a target trough blood level of 125–175 ng/mL [24]. Patients treated with rituximab received 500–1000 mg on days 1 and 15, with a possibility for a second course of rituximab after 6–12 months depending on immunological or clinical response [24].

### 2.3. Study Follow-Up and Data Collection

The follow-up visits were performed every three months in the first year and every 6 months thereafter. Baseline data were collected via electronic medical record review of patients at the time of NS diagnosis and included: age, gender, body mass index, history of smoking, presence of arterial hypertension, occurrence of thromboembolic events, serum creatinine, albumin and total proteins, lipid panel, serum fibrinogen, hemoglobin, serum IgG level, presence and titer of anti-PLA2R antibody, 24-h proteinuria and hematuria. Glomerular filtration rate (eGFR) was estimated using the Chronic Kidney Disease Epidemiology Collaboration (CKD–EPI) equation. MN stage was assessed by electron-microscopy according to Ehrenreich–Churg classification [22]. Nephrotic syndrome (NS) was defined as 24 h proteinuria over 3.5 g/day in association with hypoalbuminemia.

Anti-PLA2R antibodies were determined by an ELISA assay (EUROIMMUN AG, Lübeck, Germany). A positive serology was defined as an ELISA titer ≥ 20 RU/mL. Patients were stratified by anti-PLA2R antibody status: PLA2R-negative pMN, PLA2R-positive pMN with a low titer (<200 RU/mL) or high titer (≥200 RU/mL).

### 2.4. Study Endpoints

The primary outcomes evaluated during the follow-up period were the occurrence of immunological and clinical remission (either complete or partial remission). Complete remission (CR) was defined as proteinuria of no more than 0.3 g/day and a serum albumin of at least 3.5 g/dL. Partial remission (PR) was defined as a reduction in proteinuria of at least 50% from baseline to a level between 0.3 and 3.5 g/day [24]. Immunological remission (IR) was defined as a decrease in anti-PLA2R antibody titer below 20 RU/mL.

### 2.5. Statistical Analysis

Continuous variables were expressed as either mean ± standard deviation or median (interquartile range: 25th–75th percentiles), according to their distribution, and categorical variables as percentages. Differences between groups were assessed in case of continuous variables by Student *t* test, Mann–Whitney test, one-way ANOVA or Kruskal–Wallis test, according to the distribution of dependent variables and the level of independent variable, and, in case of categorical variables, by Pearson χ^2^ test or Fisher’s exact test. When evaluating differences between the variables related to the outcome of the study, between group differences at each follow-up time point were assessed by Student t test or by Mann–Whitney test, according to their distribution.

The cumulative proportion of complete or partial remission was assessed by Kaplan–Meyer method, and the log-rank test was used for comparisons. Univariate and multivariate Cox proportional hazard regression analyses were performed to identify independent predictors of clinical remission. The results of Cox analyses are expressed as a hazard ratio (HR) and 95% confidence interval (95%CI). In all analyses, *p*-values are two-tailed, and all *p* values less than 0.05 were considered statistically significant.

Statistical analyses were performed using the SPSS program (SPSS version 20, Chicago, IL, USA) and XLSTAT (Addinsoft 2019, XLSTAT statistical and data analysis solution. https://www.xlstat.com. Boston, MA, USA).

## 3. Results

Of the 92 patients diagnosed with pMN in our clinic, 65 patients had periodic anti-PLA2R antibody determination (every three months in the first year and every 6 months thereafter), at least 24 months of follow-up and were further included in the study.

The study cohort had a mean age of 53 ± 12 years, 71% of patients were males, and the mean eGFR was 62 ± 29 mL/min/1.73m2. The majority of patients had nephrotic syndrome (81.5%), the median level of 24-h proteinuria and the mean serum albumin were 8.7 g/d (IQR: 5.2–15.4) and 2.79 ± 0.65 g/dL, respectively. During the follow-up period, 20% of patients developed thromboembolic events. In terms of PLA2R serology, 80% of patients had anti-PLA2R antibodies at diagnosis with a median level of 199 RU/mL (IQR: 100–320), 48% of these having an anti-PLA2R antibody level over 200 RU/mL. Overall, the majority of patients received antiproteinuric treatment (97%) and an immunosuppressive regimen (92.3%). Most patients received cyclophosphamide-based regimens (47.7%), with 29.2% and 15.4% of the study cohort being treated with calcineurin inhibitors (CNI) or rituximab-based regimens, respectively, while five patients (7.7%) did not receive IS therapy. Immunological remission was achieved in 63.5% of patients with a positive serology at a median time of 18 months (IQR: 9.7–24). The renal response rate (complete and partial remission) was 73.8%, of which 32.2% of patients achieved a complete remission, and 41.5% of patients achieved a partial remission at some point during the follow-up period. The median time to complete and partial remission was 18 months (IQR: 7.5–24) and 24 months (IQR: 12–24), respectively. During the follow-up period, four patients presented a relapse of proteinuria (two patients were treated with cyclophosphamide-based regimens, one with calcineurin inhibitors and one was treatment-naïve).

Subsequently, we analyzed the immunological and clinical response according to the baseline anti-PLA2R antibody status and the treatment regimen. In the study cohort, 13 patients had a PLA2R-negative pMN, while, of those with PLA2R-associated pMN, 27 patients had a low anti-PLA2R antibody titer (<200 RU/mL), and 25 patients had a high anti-PLA2R antibody titer at baseline (≥200 RU/mL). Patients with PLA2R-negative pMN had a better renal function (eGFR 79 ± 36 mL/min), higher serum albumin (3.1 ± 0.5 g/dL) and lower 24 h proteinuria (7.2 g/day, IQR: 3.4–11.1) compared to those with PLA2R-associated pMN (Table 1).

Additionally, the clinical outcome was better in patients with PLA2R-negative pMN compared to patients with PLA2R-positive pMN. These patients had a higher percentage of complete remissions (46.2%, compared to 33.3% in those with low anti-PLA2R antibody titer or 24% in those with high anti-PLA2R antibody titer), a faster decline of 24 h proteinuria and lower time to complete remission (Table 1 and Table 2, Figure 1, Figure 2 and Figure 3).

In multivariate Cox regression analysis, patients with PLA2R-negative pMN had a 3.1-fold and a 2.87-fold higher chance for achieving a complete or partial remission compared to patients with high anti-PLA2R antibody titer or to all PLA2R-positive patients, respectively (Table 3). In terms of anti-PLA2R titer, patients with a low titer had a tendency for a milder clinical picture of pMN at diagnosis and for a better prognosis (higher percentage of immunological or clinical remissions) compared to those with a high titer (Table 1 and Table 2, Figure 1 and Figure 2). However, after multivariate adjustment, we did not identify the anti-PLA2R antibody titer as an independent predictor of clinical remission (Table 3).

We then analyzed the remission rates according to treatment regimen. In terms of baseline characteristics, patients treated with cyclophosphamide-based regimens had a worse renal function (serum creatinine, 1.76 ± 0.91 mg/dL), higher 24 h proteinuria (median 11.4 g/day, IQR: 7–17.4) and lower serum IgG level (median 470 mg/dL, IQR: 327–618) compared to patients treated with other IS regimens or without IS, consistent with a more aggressive disease (Table 4). Regarding the immunological or clinical remission, we did not identify significant differences between different therapeutic interventions. However, in the treatment-naïve group (*n* = 5), 66.7% of patients achieved a spontaneous immunological remission despite that the median level of anti-PLA2R antibody titer was similar to patients treated with different IS regimens (Table 4). Additionally, despite having a similar immunological activity, these patients had a milder clinical phenotype with a lower 24 h proteinuria compared to patients treated with different IS regimens and a clinical response rate of 100% (complete or partial remission) (Table 4 and Table 5, Figure 3). By comparison, patients treated with rituximab-based regimens had the lowest clinical response rate and the highest level of proteinuria at the last follow-up visit (Table 4 and Table 5).

In univariate and multivariate Cox regression analysis, the most important predictors of clinical remission were baseline proteinuria, the achievement of immunological remission at 24 months and the baseline negative serology (Table 3). Patients with a baseline 24-h proteinuria of less than 8 g/day, with an immunological remission at 24 months or with a PLA2R-negative pMN had a 2.4-fold, 2.2-fold and 2.87-fold, respectively, higher chance of achieving a clinical response (either complete or partial remission) (Figure 2, Table 3). Renal function at diagnosis, type of therapeutic intervention or anti-PLA2R antibody titer did not predict the occurrence of clinical remission (Table 3).

## 4. Discussion

In this study, we have shown that patients with PLA2R-negative pMN had a better prognosis with an approximately 3-fold higher chance of achieving a clinical remission (either partial or complete remission) and a faster decline of proteinuria compared to patients with PLA2R-associated pMN. Additionally, baseline proteinuria seems to be a more important predictor of clinical outcome than anti-PLA2R-ab titer.

Despite the increasing prevalence of diabetic nephropathy or focal and segmental glomerulosclerosis in certain subpopulations, primary membranous nephropathy remains an important cause of nephrotic syndrome [2]. Since the landmark papers describing, in 2009, the M-type phospholipase A2 receptor and, in 2014, the Thrombospondin type-1 Domain Containing-7A as the first human target autoantigens, the field of MN has significantly changed, and currently several new “putative” antigens have been proposed to underlie different phenotypes of MN (e.g., malignancy-associated, MN in the context of systemic autoimmune disorders) [9].

The advent of commercial assays made anti-PLA2R-ab candidate biomarkers for pMN, both from a diagnostic and a prognostic standpoint [2]. The diagnostic role has been assessed in several studies with a general agreement that anti-PLA2R-ab have a good accuracy to discriminate pMN from secondary forms or other patterns of glomerular injury [10]. Apart from this diagnostic role, anti-PLA2R-ab strongly correlate with disease activity as they have been shown to appear months to years prior to clinical manifestations, disappear prior to a clinical remission and recur prior to a relapse [17,28,29,30]. Nonetheless, given the possibility for a spontaneous remission in about a third of patients with pMN, the accurate characterization of the risk of disease progression at diagnosis is mandatory [22,31]. The threshold for anti-PLA2R-ab titer to define a high risk of progression is debatable and arbitrarily set at 150–200 RU/mL [22,32,33]. In our cohort, patients with a low titer of anti-PLA2R-ab (<200 RU/mL) had only a tendency for a higher remission rate and a faster decline of proteinuria. However, after multivariate adjustment, anti-PLA2R-ab titer was not identified as an independent predictor of clinical remission. On the other hand, patients with a negative serology had a 3-fold higher chance for achieving a clinical remission (HR, 3.11; 95%CI, 1.13–8.53). These findings suggest that PLA2R-negative pMN, possibly driven by one of the more recently described autoantigens, might represent a distinct entity. Similarly, Song et al. have shown, in a study that included 66 patients with biopsy-proven MN, that anti-PLA2R-ab-positive patients, compared to those anti-PLA2R-ab-negative, had more severe proteinuria at diagnosis, a higher incidence of chronic kidney disease and lower spontaneous remission rate [18]. In addition, other studies have shown that anti-PLA2R-ab positivity was associated with the severity of the nephrotic syndrome at diagnosis and with a worse prognosis [11,16]. A distinct clinical phenotype and outcome between antigen-positive and antigen-negative patients was also seen for other subset of MN patients. A recent study showed that patients with MN and exostosin positivity were younger, had less chronicity features on kidney biopsy and were less likely to progress to ESRD compared to exostosin-negative MN patients [34]. By converging data from these studies, the hypothesis that each distinct antigen might drive a different clinical phenotype is emerging. This might explain the overall favorable clinical outcome of PLA2R-negative pMN, the clinical scenario of these patients resembling to the exostosin positivity seen in the previous study [34]. Accordingly, the shift in MN is toward a target antigen-based classification, and this approach awaits validation [35].

In our study, antibody status at the end of follow-up, but not antibody titer at baseline, was associated with clinical remission (HR, 2.24 for immunological remission at 24 months; 95%CI, 1.04–4.85). Similar findings were seen in the study of Bech et al., in which 58% of patients that achieved an immunological remission at the end of therapy were in persistent clinical remission at 5 years, compared to none of the patients that remained immunologically active [12]. Apart from the anti-PLA2R-ab titer, we have observed that proteinuria at baseline may be a more important predictor of clinical remission in patients with pMN. In the analysis stratified by the treatment approach we did not identify significant differences between the IS regimens. Moreover, IS-naïve patients had a similar clinical outcome as the IS-treated patients. This might be explained by a different clinical phenotype or a different pathogenicity of antibodies, rather than the immunological activity of these patients. Patients that received different IS regimens and those IS-naïve had a comparable anti-PLA2R-ab titers, but IS-naïve patients had significantly lower proteinuria (median 2.8 g/day compared to 7.2–11.4 g/day in patients that received IS). Moreover, IS-naïve patients enrolled in our study would have been excluded from the most recent randomized clinical trials as the inclusion criteria required a proteinuria over 3.5 g/day (RI–CYCLO trial), over 4 g/day (STARMEN trial) or over 5 g/day (MENTOR trial) [24,25,26]. By stratifying the risk of progression solely by anti-PLA2R-ab titer, these patients would have been improperly exposed to IS therapy and its associated adverse events [36,37]. By contrary, in our cohort, these patients achieved a spontaneous immunological and clinical remission. Again, this suggests that, in the real-world clinical practice, the indication of IS treatment is still driven by the severity of the nephrotic syndrome at baseline and not by the immunological activity. This observation is sustained by the identification of the severity of baseline proteinuria as the most important predictor of clinical remission. Patients with a baseline proteinuria of less than 8 g/day had a 2.4-fold higher chance for a clinical remission compared to patients with higher levels of proteinuria.

Our study has several limitations that need to be acknowledged. First, this is a single-center, observational, non-interventional study with the treatment approach selected by the attending physician. This study nature differs significantly from a RCT setting, and this may limit the generalizability of our results. Secondly, the small number of patients from each subgroup may have prevented us from identifying significant differences according to different baseline characteristics (e.g., anti-PLA2R-ab titer, IS regimen). Third, we did not stain kidney biopsies for PLA2R antigen, and PL2AR-positive MN may have been missed in those cases with negative serology but positive antigen tissue staining. However, the study enrolled adequately and prospectively followed patients for a long period, 24 months, reflecting the real-world clinical practice, and this provides strength to our study.

## 5. Conclusions

In conclusion, we identified a different clinical phenotype between PLA2R-positive and PLA2R-negative pMN. In such cases, the pathogenesis of MN is supposedly driven by a different autoantigen, such as the recently described ones. Additionally, we have shown that baseline proteinuria seems to be a more important predictor of clinical outcome than sPLA2R-ab titer.

## Figures and Tables

**Figure 1 jcm-10-02624-f001:**
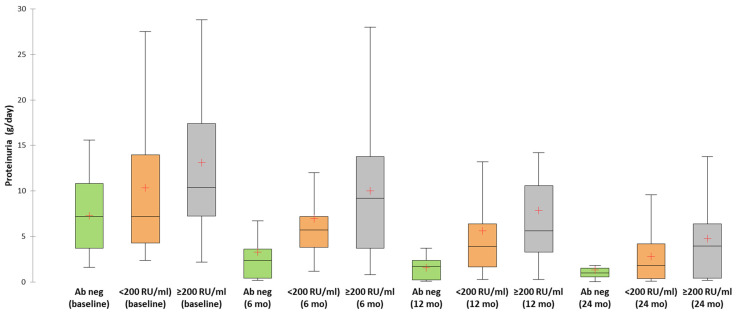
Evolution of proteinuria according to anti-PLA2R antibody status.

**Figure 2 jcm-10-02624-f002:**
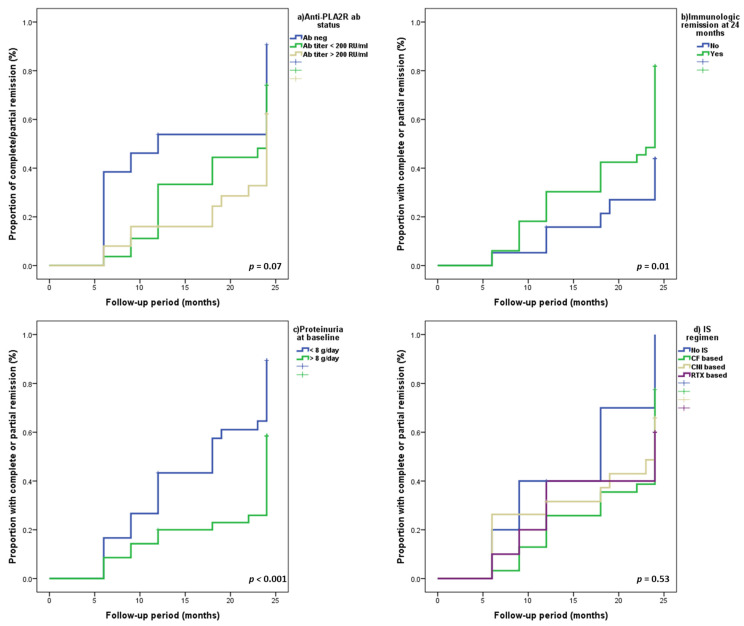
Cumulative proportion of patients with complete or partial remission according to: (**a**) PLA2R status, (**b**) immunological remission at 24 months, (**c**) baseline proteinuria and (**d**) IS regimen.

**Figure 3 jcm-10-02624-f003:**
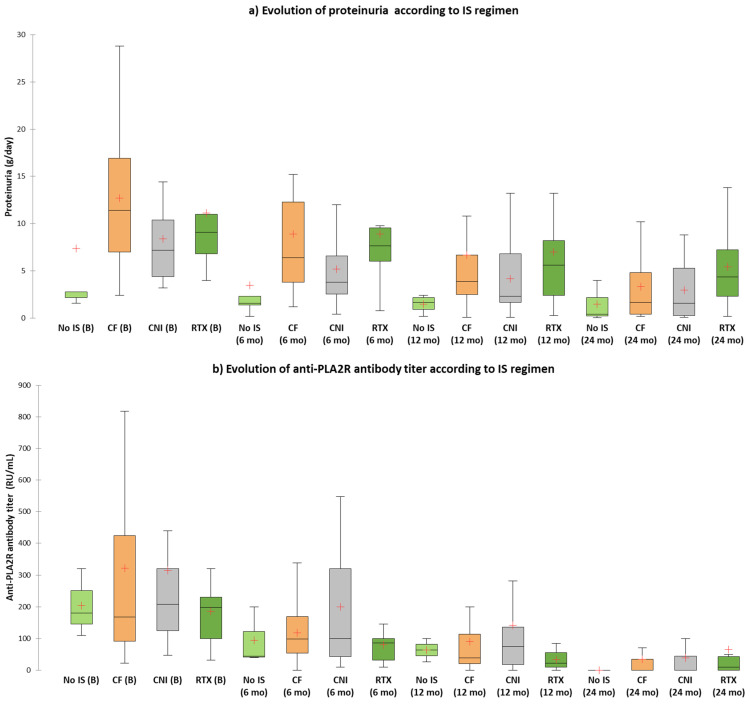
(**a**) Evolution of proteinuria according to IS regimen; (**b**) Evolution of anti-PLA2R antibody titer according to IS regimen.

**Table 1 jcm-10-02624-t001:** Univariate analysis according to baseline anti-PLA2R antibody status.

Variable	Negative PLA2R	PLA2R ab <200 RU/mL	PLA2R ab >200 RU/mL	*p* Value
Number of pts	13	27	25	
Age	54 ± 13	53 ± 13	51 ± 11	0.75
Gender (%M)	76.9%	63%	76%	0.58
BMI (kg/m^2^)	31 ± 7.7	30.2 ± 4.8	30.6 ± 4.7	0.9
HTA (%)	100%	96.3%	92%	0.51
Smoking (%)	76.9%	55.6%	60%	0.42
Tromboembolic events (%)	7.7%	25.9%	20%	0.4
Serum creatinine (mg/dL)	1.15 ± 0.73	1.63 ± 0.94	1.33 ± 0.5	0.27
eGFR (mL/min)	79 ± 36	55 ± 27	61 ± 23	**0.05**
Serum Albumin (g/dL)	3.1 ± 0.5	2.8 ± 0.5	2.5 ± 0.6	**0.014**
Total Proteins (g/dL)	5.9 ± 0.9	5.6 ± 0.8	5 ± 0.7	**0.01**
Total cholesterol (mg/dL)	263 ± 104	304 ± 107	333 ± 115	0.18
Triglycerides (mg/dL)	187 ± 90	185 ± 101	223 ± 118	0.23
Fibrinogen (mg/dL)	597 ± 159	577 ± 155	642 ± 151	0.31
Hemoglobin (g/dL)	13.8 ± 1.8	13.8 ± 2.2	13.8 ± 1.6	0.96
24 h proteinuria (g/day)	7.2 (IQR: 3.4–11.1)	7.2 (IQR:4.2–14.4)	10.4 (IQR:7.1–17.4)	0.04
Hematuria (cells/mmc)	12 (IQR:5–26)	17 (IQR:9–40)	25 (IQR:15–38)	0.11
Serum IgG level (mg/dL)	474 (IQR:380–928)	470 (IQR:340–635)	470 (IQR:284–645)	0.42
Median PLAR2R ab titer (RU/mL)	-	100 (IQR:80–150)	320 (IQR: 320–467)	**<0.001**
MN stage (% of patients)				
● I	45.4%	8.3%	10.5%	**0.008**
● II	0%	33.3%	36.8%	
● III	27.3%	50%	52.6%	
● IV	27.3%	8.3%	0%	
Clinical response (% of patients)				
● No response	7.7%	25.9%	36%	0.4
● Partial remission	46.2%	40.7%	40%	
● Complete remission	46.2%	33.3%	24%	
Time to partial remission (mo)	18 (IQR:10.5–24)	12 (IQR:12–24)	18.5 (IQR:8.25–24)	0.95
Time to complete remission (mo)	6 (IQR:6–12.7)	18 (IQR:12–23.5)	23 (IQR:15.7–24)	**0.048**
Immunological remission (% of patients)	-	70.4%	56%	**<0.001**
Time to immunological remission (mo)	-	12 (IQR:9–18)	15 (IQR:8.25–18)	0.99
Antiproteinuric therapy (% of pts.)	92.3%	100%	96%	0.39
Immunosuppressive therapy (% of pts.)				
No IS	15.4%	7.4%	4%	0.56
Cyclophosphamide-based regimens	30.8%	51.9%%	52%	
CNI-based regimens	46.2%	22.2%	28%	
Rituximab-based regimes	7.7%	18.5%	16%	

Abbreviations: M, males; BMI, body-mass index; HTA, arterial hypertension; eGFR, estimated glomerular filtration rate; MN, membranous nephropathy; IS, immunosuppression; CNI, calcineurin inhibitors; mo, months; pts, patients; IQR, interquartile range. *p* values less than 0.05 were considered statistically significant, and are marked with bold.

**Table 2 jcm-10-02624-t002:** Evolution of sPLA2R-ab titer and 24-h proteinuria according to baseline anti-PLA2R antibody status.

Period	Negative anti-PLA2R-ab		Anti-PLA2R-ab <200 RU/mL		Anti-PLA2R-ab >200 RU/mL		*p*
	Anti-PLA2R antibodies						
	Titer (IQR) (RU/mL)	Immunologic Remission (%)	Titer (IQR) (RU/mL)	Immunologic Remission (%)	Titer (IQR) (RU/mL)	Immunologic Remission (%)	
Baseline			100 (80–150)	-	320 (320–467)	-	-
3 mo			85 (52–140)	0%	235 (166–320)	0%	-
6 mo			54 (35–100)	7.4%	145 (88–260)	12%	0.66
9 mo			46 (20–100)	25.9%	82 (49–213)	16%	0.38
12 mo			32 (10–76)	40.7%	75 (13–240)	28%	0.33
18 mo			32 (0–55)	55.6%	32 (0–81)	52%	0.79
24 mo			0 (0–35)	70.4%	10 (0–61)	56%	0.28
	**Proteinuria**						
	**Level (g/day)**	**CR/PR** **(%)**	**Level (g/day)**	**CR/PR** **(%)**	**Level (g/day)**	**CR/PR** **(%)**	***p***
Baseline	7.2 (3.45–11.1)	-	7.2 (4.2–14.4)	-	10.4 (7.1–17.4)	-	-
3 mo	5.8 (1.9–8.8)	0%	6 (4.2–9.2)	0%	9.2 (6–14.9)	0%	-
6 mo	2.4 (0.4–5.1)	30.8%	5.7 (3.8–7.5)	3.7%	9.2 (3.15–14)	8%	0.03
9 mo	1.55 (0.35–4.05)	46.2%	4 (2–8.4)	11.1%	8 (5.5–10.8)	16%	0.02
12 mo	1.7 (0.17–2.5)	61.5%	3.9 (1.45–6.9)	33.3%	5.6 (2.95–10.7)	20%	0.03
18 mo	1 (0.15–1.9)	61.5%	3.2 (0.43–6.8)	44.4%	5 (1.7–10.7)	24%	0.06
24 mo	1 (0.3–1.8)	84.6%	1.8 (0.3–4.8)	74.1%	3.95 (0.4–6.52)	60%	0.25

**Table 3 jcm-10-02624-t003:** Univariate and multivariate Cox proportional hazard regression analysis regarding predictor of treatment response (partial or complete remission).

Variable	Univariate Analysis		Multivariate Analysis (Model A)		Multivariate Analysis (Model B)	
	Hazard Ratio (95% CI)	*p*-Value	Hazard Ratio (95% CI)	*p*-Value	Hazard Ratio (95% CI)	*p*-Value
Serum Creatinine (for 1 mg/dL)	1.16 (0.79–1.7)	0.44	1.21 (0.76–1.93)	0.41	1.22 (0.76–1.95)	0.4
Serum Albumin (for 1 g/dL)	1.35 (0.89–2.04)	0.15	0.86 (0.48–1.52)	0.6	0.87 (0.5–1.52)	0.64
24 h Proteinuria (<8 g/d vs. ≥8 g/d)	2.31 (1.28–4.17)	0.005	2.38 (1.18–4.76)	0.01	2.4 (1.19–4.8)	0.01
Type of IS (vs. no IS)	-	-	-	-	-	-
● CF-based regimens	0.54 (0.18–1.57)	0.26	0.89 (0.29–2.73)	0.83	0.86 (0.28–2.62)	0.79
● CNI-based regimens	0.53 (0.17–1.65)	0.27	0.89 (0.26–2.98)	0.85	0.88(0.26–2.93)	0.83
● RTX-based regimens	0.44 (0.12–1.58)	0.21	0.64 (0.17–2.43)	0.51	0.64(0.17–2.43)	0.51
Immunological remission at 24 mo	1.46 (0.81–2.65)	0.2	2.25 (1.04–4.85)	0.04	2.26(1.05–4.87)	0.03
Anti-PLA2R ab titer (vs. >200 RU/mL)	-	-	-	-	-	-
● <200 RU/mL	1.37 (0.7–2.67)	0.35	1.13 (0.55–2.33)	0.73	-	-
● Negative serology	2.15 (0.98–4.69)	0.055	3.11 (1.13–8.53)	0.02	-	-
Anti-PLA2R ab (neg vs. pos)	1.18(0.92–3.58)	0.08	-	-	2.87(1.17–7.02)	0.02

Abbreviations: IS, immunosuppression; CNI, calcineurin inhibitors; CF, cyclophosphamide; RTX, rituximab; mo, months, ab, antibody.

**Table 4 jcm-10-02624-t004:** Univariate analysis according to immunosuppressive regimen.

Variable	No IS	CF-Based Regimens	CNI-Based Regimens	RTX-Based Regimens	*p*-Value
Number of pts	5	31	19	10	
Age (y)	50 ± 10	54 ± 12	51 ± 13	52 ± 14	0.77
Gender (%M)	100%	77.4%	52.6%	60%	0.07
BMI (kg/m^2^)	33.2 ± 6.9	30.5 ± 4.6	28.7 ± 6.6	32.5 ± 3.6	0.21
HTA (%)	80%	96.8%	94.7%	100%	0.34
Smoking (%)	80%	74.2%	42.1%	50%	0.09
Tromboembolic events (%)	0%	29%	15.8%	10%	0.3
Serum creatinine (mg/dL)	1.26 ± 0.46	1.76 ± 0.91	0.98 ± 0.25	1.27 ± 0.63	0.01
eGFR (mL/min)	64 ± 18	51 ± 28	77 ± 28	67 ± 25	0.015
Serum Albumin (g/dL)	3.26 ± 0.83	2.78 ± 0.61	2.81 ± 0.55	2.58 ± 0.85	0.31
Total Proteins (g/dL)	6.08 ± 0.32	5.37 ± 0.81	5.34 ± 0.92	5.56 ± 1.24	0.38
Cholesterol (mg/dL)	272 ± 91	290 ± 102	328 ± 113	339 ± 142	0.44
Triglycerides (mg/dL)	178 ± 119	200 ± 109	205 ± 126	203 ± 37	0.59
Fibrinogen (mg/dL)	429 ± 88	630 ± 160	641 ± 151	556 ± 106	0.02
Hemoglobin (g/dL)	13.4 ± 2.3	13.5 ± 2.2	13.9 ± 1.3	14.7 ± 1.3	0.61
24 h proteinuria (g/day)	2.8 (IQR:1.9–15.4)	11.4 (IQR:7–17.4)	7.2 (4.3–10.8)	9.1 (IQR:6.03–12.8)	0.02
Hematuria (cells/mmc)	11 (IQR:3–15)	17 (IQR:9–43)	28 (IQR:13–45)	16 (IQR:7–25)	0.09
Serum IgG level (mg/dL)	961 (IQR:708–1160)	470 (IQR:327–618)	408 (IQR:305–534)	703 (IQR:320–850)	0.009
Anti-PLA2R-ab positive pts (%)	60%	87.1%	68.4%	90%	0.21
Anti-PLAR2R-ab titer (RU/mL)	180 (IQR:110–180)	168 (IQR:89–448)	208 (IQR:111–320)	198 (IQR:100–275)	0.91
PLA2R ab titer (%)					
● <200 RU/mL	66.7%	51.9%	46.2%	55.6%	0.68
● >200 RU/mL	33.3%	48.1%	53.8%	44.4%	
MN stage (% of pts)					
● I	20%	8%	25%	25%	0.82
● II	40%	28%	18.8%	37.5%	
● III	40%	52%	43.8%	37.5%	
● IV	0%	12%	12.5%	0%	
Evolution characteristics					
Clinical response (% of pts)					
● No response	0%	22.6%	31.6%	40%	0.5
● PR	60%	41.9%	31.6%	50%	
● CR	40%	35.5%	36.8%	10%	
Time to PR (mo)	9 (IQR:9–12)	24 (IQR:12–24)	15.5 (IQR:6–24)	12 (IQR:9–24)	0.43
Time to CR (mo)	15 (IQR:6–15)	18 (IQR:12–24)	12 (IQR:6–23)	9 (IQR:9–9)	0.51
IR (% of patients)	66.7%	63%	61.5%	66.7%	0.6
Time to IR (mo)	18 (IQR:9–18)	18 (IQR:12–24)	18 (IQR:9–24)	12 (IQR:7.5–24)	0.83
Antiproteinuric therapy (%)	100%	96.8%	94.7%	100%	0.85

Abbreviations: M, males; BMI, body-mass index; HTA, arterial hypertension; eGFR, estimated glomerular filtration rate; MN, membranous nephropathy; IS, immunosuppression; CNI, calcineurin inhibitors; RTX, rituximab; CF, cyclophosphamide; mo, months; pts, patients; IQR, interquartile range; PR, partial remission; CR, complete remission; IR, immunological remission. *P* values less than 0.05 were considered statistically significant, and are marked with bold.

**Table 5 jcm-10-02624-t005:** Evolution of anti-PLA2R antibody titer and 24 h proteinuria according to immunosuppressive regimen.

Period	No IS		CF-Based Regimens		CNI-Based Regimens		RTX-Based Regimens		*p-*Value
	Anti-PLA2R Antibodies								
	Titer (IQR) (RU/mL)	IR (%)	Titer (IQR) (RU/mL)	IR (%)	Titer (IQR) (RU/mL)	IR (%)	Titer (IQR) (RU/mL)	IR (%)	
Baseline	180 (110–180)	–	168 (89–448)	–	208 (111–320)	–	198 (100–275)	–	–
3 mo	85 (51–85)	0%	140 (75–243)	0%	170 (73–317)	0%	160 (87–205)	0%	–
6 mo	44 (40–44)	0%	98 (53–170)	11.%	100 (34–320)	7.7%	86 (25–122)	11.1%	0.92
9 mo	48 (22–48)	33.3%	50 (38–100)	14.8%	110 (27–293)	30.8%	26 (10–148)	22.2%	0.65
12 mo	63 (27–63)	33.3%	39 (18–127)	33.3%	75 (0–160)	30.8%	22 (10–55)	44.4%	0.92
18 mo	24 (16–24)	33.3%	32 (0–74)	59.3%	42 (0–85)	46.2%	26 (0–60)	55.6%	0.76
24 mo	15 (10–25)	66.7%	0 (0–45)	63%	0 (0–56)	61.5%	10 (0–117)	66.7%	0.99
	**Proteinuria**								***p*-Value**
	**Level** **(g/day)**	**CR/PR (%)**	**Level** **(g/day)**	**CR/PR (%)**	**Level** **(g/day)**	**CR/PR (%)**	**Level** **(g/day)**	**CR/PR (%)**	
Baseline	2.8 (1.9–15.4)	–	11.4 (7–17.4)	–	7.2 (4.3–10.8)	–	9.1 (6.03–12.8)	–	–
3 mo	1.9 (0.65–17.1)	0%	7.4 (5.2–13.6)	0%	6.4 (3.3–9.6)	0%	9 (5.67–12.3)	0%	–
6 mo	1.6 (0.8–7.1)	20%	6.4 (3.8–12.4)	3.2%	3.8 (2.4–6.6)	21.1%	7.65 (5–12.05)	10%	0.2
9 mo	1.8 (0.9–3.15)	40%	6 (2–8.4)	12.9%	4.85 (2.07–9.8)	26.3%	8.2 (2.8–14.5)	20%	0.44
12 mo	1.65 (0.45–2.32)	60%	3.9 (2.37–7.14)	25.8%	2.3 (1.22–7.6)	36.8%	5.6 (2.4–10.7)	40%	0.44
18 mo	1.7 (0.47–2.02)	80%	3.06 (1–9.5)	35.5%	2.2 (0.6–6.1)	36.8%	5.8 (3.65–11.5)	40%	0.29
24 mo	0.4 (0.03–0.4)	100%	1.65 (0.4–4.8)	77.4%	1.6 (0.3–5.75)	63.2%	4.35 (1.4–9.45)	60%	0.58

## Data Availability

The data presented in this study are available in the article “Clinical Phenotypes and Predictors of Remission in Primary Membranous Nephropathy”.

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
