# Peer review of "Clinical Phenotypes and Predictors of Remission in Primary Membranous Nephropathy"

_jcm, 2021, doi:10.3390/jcm10122624_

Round 1
Reviewer 1 Report
In my opinion, authors should include results of PLA2R staining in renal tissue. It tells us more about disease activity than solely serum PLA2R antibodies. The glomerulopathy activity is not only dependent on serum antibodies but also on the local injuries in tissue.
Author Response
We are submitting the reply to the composition comments you have made on our manuscript entitled “Clinical phenotypes and predictors of remission in primary membranous nephropathy” coauthored by Roxana Jurubiță, Bogdan ObriÈ™că, Bogdan Sorohan, Camelia Achim, Georgia Elena Micu, Gabriel Mircescu and Gener Ismail.,
We have revised the manuscript based on the comments made by the reviewers.
Together with revised manuscript here is our answer to the reviewer’s comments.
Reviewer 1
In my opinion, authors should include results of PLA2R staining in renal tissue. It tells us more about disease activity than solely serum PLA2R antibodies. The glomerulopathy activity is not only dependent on serum antibodies but also on the local injuries in tissue.
Response: Thank you for the comments and suggestions. We did not have the possibility to stain for PLA2R antigen. We added this under the limitations of the study. Although this may be an issue for those patients with negative serology, but positive antigen staining, in our cohort the patients were monitored actively for 24 months without any signs of further antibody positivity. This is further supported by the patient that was antibody negative, received no immunosuppression and had a relapse of proteinuria without any detectable antibodies. Such cases represent true non-PLA2R-mediated MN, possibly related to other antigens.
We hope that we have addressed all the issues of your comments.
Sincerely Yours,
Bogdan Obrisca MD
Corresponding author: Bogdan Obrisca MD – Department of Nephrology, Fundeni Clinical Institute,
258 Fundeni Street, District 2, Bucharest, Romania, zip code 022328; obriscabogdan@yahoo.com@yahoo.com
Reviewer 2 Report
This is an interesting study that describes the clinical evolution of 65 patients with membranous nephropathy according to serological status. I consider it to be a well conducted and analyzed study with useful results from a clinical point of view.
Some elements to assess would be:
- Sample size somewhat reduced.
- The study determines that patients with anti-PLA2R had a better renal evolution. Nothing is mentioned about the patient's evolution, whether other antibodies were determined, and whether secondary causes of membranous nephropathy were found.
- What explanation do the authors find that the basal antibody titer did not correlate with evolution?
- The patients who received rituximab had more proteinuria, was there a positive correlation with the antibody titer? Should rituximab be used in patients with anti-PLA2R> 200 RU / ml?
- Lastly, the authors do not make any reference to the recurrence rate in their sample.
Author Response
We are submitting the reply to the composition comments you have made on our manuscript entitled “Clinical phenotypes and predictors of remission in primary membranous nephropathy” coauthored by Roxana Jurubiță, Bogdan ObriÈ™că, Bogdan Sorohan, Camelia Achim, Georgia Elena Micu, Gabriel Mircescu and Gener Ismail.,
We have revised the manuscript based on the comments made by the reviewers.
Together with revised manuscript here is our answer to the reviewer’s comments.
Revierwer 2
This is an interesting study that describes the clinical evolution of 65 patients with membranous nephropathy according to serological status. I consider it to be a well conducted and analyzed study with useful results from a clinical point of view.
Some elements to assess would be:
- Sample size somewhat reduced.
Response: We acknowledged that this is a small study under the limitations of the study.
- The study determines that patients with anti-PLA2R had a better renal evolution. Nothing is mentioned about the patient's evolution, whether other antibodies were determined, and whether secondary causes of membranous nephropathy were found.
Response: Actually, those that were anti-PLA2R negative had a better clinical remission. We have discussed that in such situation other antibodies may be involved in disease pathogenesis. However, as the test for antibodies (other than anti-PLA2R or anti-THSD7A) have not entered into worldwide clinical practice, our results are hypothesis-generator. It will be interesting to evaluate in future studies the clinical and evolution distinction between different MN-antigen driven phenotypes. Additionally, in our cohort, no patient had a THSD7A—associated MN.
- What explanation do the authors find that the basal antibody titer did not correlate with evolution?
Response: As we have suggested in the discussion section we think that proteinuria may be a more important predictor for clinical remission than anti-PLA2R antibody titer. This is further suggested by our treatment-naïve subgroup, that had a similar PLA2R titer compared to patients treated with IS, but significantly less proteinuria. This may be due to a different pathogenical strength of such antibodies, rather than the absolute value per se.
- The patients who received rituximab had more proteinuria, was there a positive correlation with the antibody titer? Should rituximab be used in patients with anti-PLA2R> 200 RU / ml?
Response: Given the small number of patients in each subgroup of therapy and the non-randomized nature of the study we cannot draw firm conclusions about which IS regimen should be administered according to different clinical scenarios. In our study, patients that received CF-based regimens had the most severe clinical phenotype. However, we did not identify significant differences between different treatment approaches in terms of immunological or clinical remission rates or time to remission.
- Lastly, the authors do not make any reference to the recurrence rate in their sample.
Response: We have added details about the recurrences.
We hope that we have addressed all the issues of your comments.
Sincerely Yours,
Bogdan Obrisca MD
Corresponding author: Bogdan Obrisca MD – Department of Nephrology, Fundeni Clinical Institute,
258 Fundeni Street, District 2, Bucharest, Romania, zip code 022328; obriscabogdan@yahoo.com@yahoo.com
Round 2
Reviewer 1 Report
I have no other remarks.